# A Bibliometric Analysis on Dengue Outbreaks in Tropical and Sub-Tropical Climates Worldwide Since 1950

**DOI:** 10.3390/ijerph18063197

**Published:** 2021-03-19

**Authors:** Shin-Yueh Liu, Tsair-Wei Chien, Ting-Ya Yang, Yu-Tsen Yeh, Willy Chou, Julie Chi Chow

**Affiliations:** 1Department of Endocrinology, Chi Mei Chiali Hospital, Tainan 700, Taiwan; yuehls@gmail.com; 2Department of Medical Research, Chi-Mei Center, Tainan 700, Taiwan; smile@mail.chimei.org.tw; 3Medical Education Center, Chi Mei Medical Center, Tainan 700, Taiwan; u102001309@cmu.edu.tw; 4School of Medicine, College of Medicine, China Medical University, Taichung 400, Taiwan; 5Medical School, St. George’s University of London, London SW17 0RE, UK; jess97yeh@gmail.com; 6Department of Physical Medicine and Rehabilitation, Chi Mei Medical Center, Tainan 700, Taiwan; 7Department of Physical Medicine and Rehabilitation, Chung San Medical University Hospital, Taichung 400, Taiwan; 8Department of Paediatrics, Chi Mei Medical Center, Tainan 700, Taiwan; 9Department of Pediatrics, School of Medicine, College of Medicine, Taipei Medical University, Taipei 100, Taiwan

**Keywords:** dengue outbreak, Pubmed Central, choropleth map, tropical area, bibliometric analysis

## Abstract

Severe dengue outbreaks (DOs) affect the majority of Asian and Latin American countries. Whether all DOs always occurred in sub-tropical and tropical areas (STTA) has not been verified. We downloaded abstracts by searching keywords “dengue (MeSH Major Topic)” from Pubmed Central since 1950, including three collections: country names in abstracts (CNA), no abstracts (WA), and no country names in abstracts (Non-CNA). Visualizations were created to present the DOs across countries/areas in STTA. The percentages of mentioned country names and authors’ countries in STTA were computed on the CNA and Non-CNA bases. The social network analysis was applied to highlight the most cited articles and countries. We found that (1) three collections are 3427 (25.48%), 3137 (23.33%), and 6884 (51.19%) in CNA, WA, and Non-CNA, respectively; (2) the percentages of 94.3% and 79.9% were found in the CNA and Non-CNA groups; (3) the most mentioned country in abstracts were India, Thailand, and Brazil; (4) most authors in the Non-CNA collections were from the United States, Brazil, and China; (5) the most cited article (PMID = 23563266) authored by Bhatt et al. had 2604 citations since 2013. Our findings provide in-depth insights into the DO knowledge. The research approaches are recommended for authors in research on other infectious diseases in the future, not just limited to the DO topic.

## 1. Introduction

Dengue is a mosquito-borne viral infection [1]. The infection causes a flu-like illness and occasionally develops into a potentially lethal complication called severe dengue [2]. Since severe dengue was first recognized in the 1950s during dengue epidemics in the Philippines and Thailand [3,4], the incidence of dengue has grown dramatically around the world in recent decades [5,6]. A vast majority of cases are asymptomatic, and hence, the actual numbers are underreported, and many instances are misclassified [7,8].

Dengue is found in subtropical and tropical areas (STTA) worldwide, mostly in urban and semi-urban areas [2]. The 2014 dengue outbreak (DO) in Tokyo was notable as it was the first time in 70 years that Japan had experienced an autochthonous transmission [9]. As such, the Japanese government is concerned about the DO and takes extra precaution against emerging infectious threats during the 2020 (or 2021) Summer Olympics and Paralympics in Tokyo [10]. Japan is partially located in the sub-tropical climate, not to mention other countries with their whole territory in sub-tropical and tropical climates. We were thus motivated to investigate the percentage of DOs that exist in STTA.

### 1.1. Dengue Worldwide in Sub-/Tropical Climates

Before 1970, only nine countries had experienced severe dengue epidemics [2]. The disease is now endemic in more than 100 countries, including countries in Africa, the Eastern Mediterranean, the Americas, Southeast Asia, and the Western Pacific. The latter three are the most seriously affected: cases exceeded 1.2 million in 2008 and over 3.34 million in 2016 [2]. Nonetheless, an underestimate of 282 times the number of officials who reported dengue cases was addressed in India for one district [8].

Traditionally, combating communicable diseases depends on surveillance, preventive measures, outbreak investigation, and the establishment of control mechanisms. The authors [11] applied text-mining cluster analysis for monitoring trends discussing dengue cases and detecting peaks in reported cases. We doubted whether the percentage of DOs in STTA could be obtained via capturing DO publications by computing the formula (=*n*/*N*; *n* = the number of documented countries with DO in STTA, *N* = the total number of DO countries in documents) instead of using the traditional surveillance, outbreak investigation, or other measures. Consequently, we were interested in conducting bibliometric analyses to estimate the DO percentage in STTA.

### 1.2. Using Publication Sources Reporting DO Areas

If any DO were reported in a country/area, the authors of papers on it would share their insights and experiences with their peers. The name of countries/areas would be (and must be) addressed in their abstracts. Accordingly, the DO areas can be obtained via the text-mining technique [12,13]. Thus, searching country nouns in DO article abstracts for extracting the DO areas worldwide might be viable, particularly for understanding the percentage of DO in STTA. If articles were not directly related to the DO, the country names were usually absent or only contained adjectives instead of nouns in the abstract, such as experimental studies or issues on the Japanese encephalitis virus [14].

### 1.3. The Definition of Sub-Tropical and Topical Climates

The tropics have been historically defined as lying between the Tropics of Cancer and Capricorn, located at latitudes 23.5° north and south, respectively [15]. The poleward fringe of the subtropics is thus located at latitudes approximately 35° north and south, respectively [14]. A country (or area) totally or partially located in the regions between +/− 35° latitudes was defined as STTA (e.g., the United States and Japan). The percentage of DO in STTA was calculated using the formula mentioned in Section 1.1.

### 1.4. The Concerns and Research Questions in This Study

Apart from the percentage of DO in STTA, the following five important questions were also required for further investigations: (1) which countries/areas are frequently mentioned and affiliated by authors in DO articles; (2) what articles with DO content that mentions country names beyond STTA are often addressed and discussed in the literature; (3) which affiliated countries contribute most to the DO academic literature based on publications and citations; (4) which DO articles were cited most and what research types are featured; and (5) what are the differences in research characteristics between DO articles with country names in abstracts (CNA) and without country names in abstracts (Non-CNA).

### 1.5. Objectives

The percentage of DO in STTA based on CNA and Non-CNA publications was computed through text-mining techniques. Five tasks were followed to report the research questions mentioned above, including (1) the prominent countries/areas mentioned in DO articles; (2) the DO content in affiliated countries beyond the STTA; (3) the countries with the most contributions to the dengue discipline; (4) the most cited articles and features in research types; and (5) the comparison of differences in research characteristics between the two CNA and Non-CNA collections.

## 2. Materials and Methods

### 2.1. Data Source

We downloaded article abstracts by searching keywords “dengue [MeSH Major Topic]” from January 1950 to December 2020 in PubMed Central (PMC) and identified those abstracts that included country names (CNA), those without abstracts (WA), and those without country names in abstracts (Non-CNA). Only the two CNA and Non-CNA collections were involved in this study. The CNA articles were identified by using text-mining techniques with a loop in search of any country noun in the abstract; see Section A.1 and Section A.2.

All data used in this study were downloaded from PMC, which means the study did not require ethical approval according to the regulations of the Taiwan Ministry of Health and Welfare.

### 2.2. Task 1: The Most Prominent and Productive Countries/Areas Shown on Choropleth Maps

The most prominent and productive countries/areas were displayed on choropleth maps [16,17,18] in terms of CNA and Non-CNA collections. Countries/regions in CNA imply the occurrence of DO released in the article. In contrast, authors in Non-CNA indicate the contributions to DO disciplines using the bibliometric metric to evaluate.

The percentage of countries/areas located within the STTA were also observed and compared based on the two CNA and Non-CNA collections.

### 2.3. Task 2: The DO Contents in Affiliated Countries beyond the STTA

A pyramid plot [19] was applied to display the number of countries mentioned in abstracts using the CNA collections. For example, New Zealand and Finland were the farthest south and the farthest north countries listed in the CNA sample. Several typical DO documents would be explained why their country names outside the STTA were involved in the CNA group.

### 2.4. Task 3: The Countries with the Most Contributions to the DO

The top-cited articles with at least 100 citations (called T100DO) were extracted from the two CNA and Non-CNA collections; see Section A.3. We quantified country research contributions based on citations and author positions in the article byline using an author-weighted scheme (AWS) [20,21] for computing the research contributions to DO. The first author earned the most credit (around 63%), followed by corresponding authors (deemed as the last author with around 12% credit), and other middle authors with decreasing credits shared in the remainder weights. All summed weighted credits equaled 1.0 in an article.

The x-index [22] (=√(max*f*(I × ci)) was applied to compute the research contributions and referred to the maximal rectangle by multiplying the article citation and the corresponding publication number at i; see the two equations below:(1)$$C_{ci}=\overset{m-1}{\underset{m=0}{\sum }}(W_{c\in m}\times c_{i})$$
(2)$$x_{ci}=\sqrt{\max\left(i\times c_{i}\right)}$$
where *m* − 1 denotes the number of authors (i.e., countries in this study) in an article, see Equation (2). C*_ci_* is the country-weighted citations. *W* represents the weight according to the author’s order [20,21] in the article. The x-index is determined by the root product of publication at *i* and the corresponding citations (*ci*) when article citations are sorted in descending order; see Equation (2).

Finally, the top three most cited countries/areas were highlighted on a choropleth map. The darker the color, the higher the x-index was for the country. It is worth noting that the US’s states and China’s provinces/metropolitan cities/areas were together compared in x-indexes with other countries/areas. Otherwise, the US and China might always have dominated the research contributions in a discipline.

### 2.5. Task 4: The Most Cited Articles and Feature in Research Types

Article types based on medical subject headings (MeSH terms) were clustered by performing a social network analysis (SNA) [18,19]. In keeping with the Pajek guidelines [23] using SNA, we defined any entity (e.g., country/MeSH term/article with the PMID code) as a node (or actor/vertex) connected through the edge of the line. The AWS [11,20] was combined with the SNA to highlight the articles with the most weights (around 63%) as the first author placed at the first position in an article byline, followed by other entities (e.g., MeSH terms, affiliated countries, and publication years).

A node was sized by the weights, which were related to (1) the number of connections between two nodes [20,21], and (2) the weighted citations based on Equation (1). As such, countries (or an article) with exceptionally higher citations were highlighted with bigger bubbles sized by the weights. Articles with citations were linked to the corresponding (1) author-affiliated country and (2) article types denoted by MeSH terms in a network.

The absolute advantage coefficient (AAC in Equation (3)) [24,25,26,27,28] was applied to quantify the domain concentration based on the shares in a given community or membership, such as the concentration ratio [29] of companies’ market shares in a given industry.
(3)$$\mathrm{AAC}=\left(\frac{\frac{\gamma_{1}}{\gamma_{2}}}{\frac{\gamma_{2}}{\gamma_{3}}}\right)÷\left[1+\left(\frac{\frac{\gamma_{1}}{\gamma_{2}}}{\frac{\gamma_{2}}{\gamma_{3}}}\right)\right],$$
where AAC, ranging between 0.0 and 1.0, is determined by the citations in the top three most cited articles denoted by $\gamma_{1}$, $\gamma_{2}$, and $\gamma_{3}$. The higher ACC for the top 1 article means more strengthened momentum against the other next two counterparts.

### 2.6. Comparison of Features between CNA and Non-CNA Samples

We made a Microsoft Excel (Microsoft Corp, Albuquerque, N.M., United States) VBA (visual basic for application) module to handle the data and draw a forest plot [30] for comparing the features of CNA and Non-CNA articles based on the proportions of MeSH terms with the most centrality degree (CD) in SNA mentioned in the previous section. In addition, the standardized mean differences (SMD) in citations between the two CNA and Non-CNA groups were examined and compared as well.

### 2.7. Statistical Tools and Data Analysis

The weighted CDs for each actor (or node) and the corresponding partitioned clusters were analyzed by author-made Excel modules based on the SNA algorithms in Pajek; see Section A.2. We created pages of Hypertext Markup Language (HTML) used to display on Google Maps [31]. All relevant information was linked to dashboards. For instance, the article immediately appears on the PMC website when clicking the bubble of the article of interest on the dashboard. The study flowchart is shown in Figure 1.

## 3. Results

### 3.1. The Two CNA and Non-CNAsamples Used in This Study

A total of 13,449 abstracts were extracted from the PMC, including 3427 (25.48%), 3137 (23.33%), and 6884 (51.19%) in the CNA, Non-abstract, and Non-CNA groups, respectively.

In total, 170,975 citations were matched to the 10,311 citable papers in the two collections (i.e., 30.5% vs. 60.1% with corresponding impact factor (IF) of 15.91 and 16.91 for CNA and Non-CNA, respectively); see details in Section A.2. The T100DO articles are collected and listed in Section A.3.

### 3.2. Task 1: The Most Prominent and Productive Countries/Areas Shown on Choropleth Maps

In Figure 2, we observed that the country names in abstracts accounted for 94.3% (4157/4419) in STTA. About 79.85% (9197/11,517) of authors were from the STTA. The most frequently mentioned counties in abstracts were India, Thailand, and Brazil. The US, Brazil, and China had the most authors in the Non-CNA group.

It is worth noting that the consistently matched number rate was 0.54 = 1874/3427, calculated in the CNA sample, indicating half of the authors who reported the DO evolution along with their own country names, which was different from the Non-CNA articles irrelevant to the DO, but merely with the dengue disease or event. It is because most DO-related articles should (and must) include country/area names in the abstract, like the earthquake articles always reporting the damage location (or country/area) in their articles.

### 3.3. Task 2: The DO Contents in Affiliated Countries beyond the STTA

About 94.3% (4157/4419) of countries/areas were mentioned in article abstracts, shown in Figure 3 based on the 3427 articles in the CNA collection.

Several typical DO documents that were published outside the sub-/tropics are listed in Table 1. We can see that the countries mentioned in abstracts are not attributed to the DO but to the patients who traveled from other DO countries/areas in the STTA. That is, many cases were from travelers infected in Southeast Asia returning to the targeted regions.

### 3.4. Task 3: The Countries with the Most Contributions to the DO

The most cited countries/areas were Brazil, the United Kingdom, and the State of Maryland in the US, based on the T100DO articles (see Figure 4 and Section A.3), indicating that those countries were the ones with most contributions to the DO literature since 1950.

### 3.5. Task 4: The Most Cited Articles and Feature in Research Types

The most cited article (PMID = 23563266 with 2604 citations in PMC) [5], entitled “The Global Distribution and Burden of Dengue”, was published by the journal *Nature* in 2013 and authored by Bhatt et al. from the UK; see the biggest black bubble at the middle in Figure 5. The AAC was 0.77 (=(3.27)/(1 + 3.27)).

Other highly cited articles were authored by (1) Gubler (US, with 796 citations published in 1998) [42], (2) Nafeev (Russia, 796 citations, published in 2011) [43], and (3) Van-Mai Cao-Lormeau et al. (France, 766 citations, published in 2016) [44], see Figure 5 and Section A.3. Readers are invited to scan the QR code in Figure 5 and click on the black bubble of the article to read the document in PMC.

Three article types (i.e., represented by epidemiology, immunology, and pharmacology) are clustered in Figure 5, based on MeSH terms on the T100DO articles by performing the SNA (see the MP4 video in Section A.1).

### 3.6. Task 5: Comparison of Features between CNA and Non-CNA Samples

The top 14 MeSH terms with the higher weighted CD were extracted from Figure 5. Only three MeSH terms (i.e., isolation and purification, pharmacology, and drug effects) presented significantly different sizes in the two CNA and Non-CNA samples shown in Figure 6, indicating that the CNA articles had a high proportion of the three MeSH terms when compared to the Non-CNA sample. Furthermore, no difference was found in research achievements denoted by the x-indexes between the two samples, but the three (*p* < 0.05) in red color; see the bottom in Figure 6. It is worth mentioning that the comparison was made merely based on the T100DO articles.

### 3.7. Online Dashboards Shown on Google Maps

All of the visualizations in figures were displayed on dashboards. Readers are invited to click on the links [45,46,47,48,49] to further examine the information on Google Maps.

## 4. Discussion

### 4.1. Principle Findings

We found that: (1) The percentage of the CNA sample that occurred in STTA was 94.3% (4157/4419), though some countries (e.g., the US and Japan) with partial territories within the sub-tropical climate were classified into the STTA. (2) The names most mentioned in abstracts were those of researchers from India, Thailand, and Brazil, indicating that they are the countries with DO that are most frequently discussed in academia. (3) The percentage of the Non-CNA authors based in STTA was 79.9%, implying that nearly 20% of co-authors were affiliated with countries beyond the STTA. (4) The US, Brazil, and China contributed the most articles to the DO disciple, meaning that adequate grant funding and resources on DO were invested in those three countries. (5) The most cited article (PMID = 23563266, having 2604 citations since 2013) was authored by Bhatt et al. from the UK. The AAC of the article was 0.77, indicating a strongly absolute advantage over the next two highly cited articles on the dengue topic in the past. (6) Only three MeSH terms (i.e., isolation and purification, pharmacology, and drug effects) were significantly different between the CNA and Non-CNA collections, implying that more country names in these three MeSH terms are likely involved in abstracts.

### 4.2. What This Study Contributes to Current Knowledge

Dengue is listed as a “neglected tropical disease” (NTD). The NTD has the following main features: (i) poverty-related, (ii) endemic to the subtropics and tropics, (iii) lacking public health attention, (iv) having poor research funding and shortcomings in R&D, (v) usually associated with high morbidity but low mortality, and (vi) often having no specific treatment available [50]. Dengue meets most of these criteria, but not all.

The one second criterion—that of dengue being endemic to STTA—has been addressed in many articles [51,52,53]. The percentage of DOs in STTA has remained unknown until now. Not only are a vast majority of dengue cases asymptomatic, leading to the actual numbers of dengue cases being underreported [2,7,8], but also many cases are misclassified [2]. Based on the publications in PMC, we verified that about 94.3% of dengue areas are located in STTA.

Referring to the bottom panel in Figure 2, we see many countries in STTA without any article published in PMC, such as the Lao People’s Democratic Republic and many in Africa. Lacking research funding for dengue in those areas might decrease the percentage of published articles on DO in STTA. Although research funding for dengue has increased exponentially in the past two decades, much was invested in the area of dengue vaccine development [51].

Another finding in this study is about the moderate rate of consistency (54%; 1874/3427) in outbreak areas and the authors’ affiliated countries in CNA and Non-CNA groups. One half of the authors were affiliated with countries that were not the same as those countries in which they reported on dengue outbreaks. Among them, about 45% of Non-CNA T100DO were first authors from the US and the UK, indicating that scholars in countries with poor research funding and shortcomings in R&D could perhaps be aided in dengue research by those scholars in industrialized nations. An example is that the most cited article (PMID = 10608744) with 562 citations [54] was authored by Vaughn from the United States, but the DO was found in Thailand.

Many publications outside the STTA were mostly attributable to issues indirectly related to DOs, such as (1) travelers infected in dengue areas, (2) dengue affected by the climate change [38,55,56,57,58,59,60], and (3) the dengue treatment development [61,62,63].

### 4.3. What Is the Strength of This Study

There are several strong features in this study. First, the text-mining technique was applied to extract the names of countries/areas in abstracts. That technique is similar to the investigation of earthquakes or worldwide epidemics. A close relationship simultaneously exists between two attributes routinely captured in abstracts—namely, the country and the topic (e.g., earthquake or the DO). Dengue fever surveillance using text mining in public media [8] (e.g., in web-accessible information sources such as discussion sites, disease reporting networks, and news outlets [64,65,66] or HealthMap) is worth performing for future outbreak detection, that is, creating a real-time surveillance system and updating news on new and ongoing disease outbreaks.

Second, choropleth maps were applied to display the distribution of publications by countries about DOs. The most famous illustration of choropleth maps was applied to the results of the 2000 US presidential election [67]. Recently, many examples of disparities in health outcomes across areas, such as DOs [68,69], disease hotspots [70], and the Global Health Observatory (GHO) maps on major health topics [71], have been presented.

Third, social network analysis [72,73,74] was applied to visualize the most cited countries/areas and articles worldwide on DOs, which is easy-to-read, easy-to-understand, and a topic of interest, yet one that has been rarely observed in the past, particularly when it comes to incorporating Google Maps with a dashboard to present study results. For instance, in Figure 5, readers can easily click on the bubble linked to PMC and immediately read the online article.

Fourth, a novel approach for plotting the forest plots is provided in the reference [46] and the abstract video in Section A.1, which are easily and clearly displayed and interpreted on dashboards laid out on Google Maps. The online forest plot can be applied to any two-pair comparison with two observed events and nonevent counts [30]. The method has been used in numerous articles related to meta-analysis in the literature [75].

Fifth, numerous bibliometric analyses on dengue fever [76,77,78,79] have disclosed the dominant countries publishing on the topic, mainly the US and China. However, our study not only involves the US states and China’s provinces/metropolitan cities/areas but also compares their quantity of abstracts with that of other countries, as shown on the choropleth map (Figure 5)—A presentation that is modern and innovative and never before seen in bibliometric analysis.

### 4.4. Limitations and Suggestions

Despite the findings shown above, several potential limitations require further research efforts in the future. First, the sample of this study only comprised articles in PMC. It should not be generalized to other libraries such as the Scientific Citation Index (Thomson Reuters, New York, NY, USA) and Scopus (Elsevier, Amsterdam, The Netherlands). As such, the most cited articles and countries are barely determined by the publications indexed in PMC.

Second, there might be some biases when text mining country names from abstracts. For example, the abstract [80] entitled “Paediatric dengue infection in Cirebon, Indonesia” was excluded from this study because the country name of Indonesia was not found in its abstract. The searching scheme requires including both title and abstract in the future.

Third, we recommend using SNA to partition clusters. SNA is not limited to the Pajek guidelines used in this study because many such kinds of software are in use in academia. The style of the visual representations might be somewhat different, but the principle and algorithm of partitions for cluster analysis are similar.

Fourth, although our suggestions are limited to DOs, other types of topics can be applied in the future, such as earthquakes, flu-like outbreaks, HIV/AIDS, and avian influenza, all of which can be mimicked using the text-mining technique in extracting strings or topics of interest from article abstracts. Details about the script in Excel are deposited in the link [31] and Section A.2.

Finally, numerous scientometrics were used for evaluating the authors’ individual research achievements (IRAs). We merely applied the x-index to illustrate the influential areas in the DO field shown in Figure 4. Authors are familiar with indices, such as h-index [81], g-/AG-index [82], and author factor impact [83], that can be used to measure countries in terms of IRAs on other topics in the future.

Furthermore, many countries did not pay attention to providing publications with reports of their dengue disease severity, leading to some bias in our study results. Hopefully, this research on the computation of DO percentage in STTA drives more motivation for authors to report their DO and enrich the literature on dengue outbreaks in the future because the association between dengue and economic growth is obvious. One study [84] on the impact of dengue on economic growth shows that the percentage of reduction of the average income per capita is 0.26% due to a dengue outbreak.

## 5. Conclusions

Not all DOs were found in STTA, but an extremely high 94.3% was found in the CNA collections, though some countries (e.g., the US and Japan) with partial territories within the STTA were included in the percentage. Our findings may provide readers with overall knowledge on the topic of dengue fever. The research methods used in this study have the potential to be applied to other infectious diseases, not just limited to the topic of dengue fever.

## Figures and Tables

**Figure 1 ijerph-18-03197-f001:**
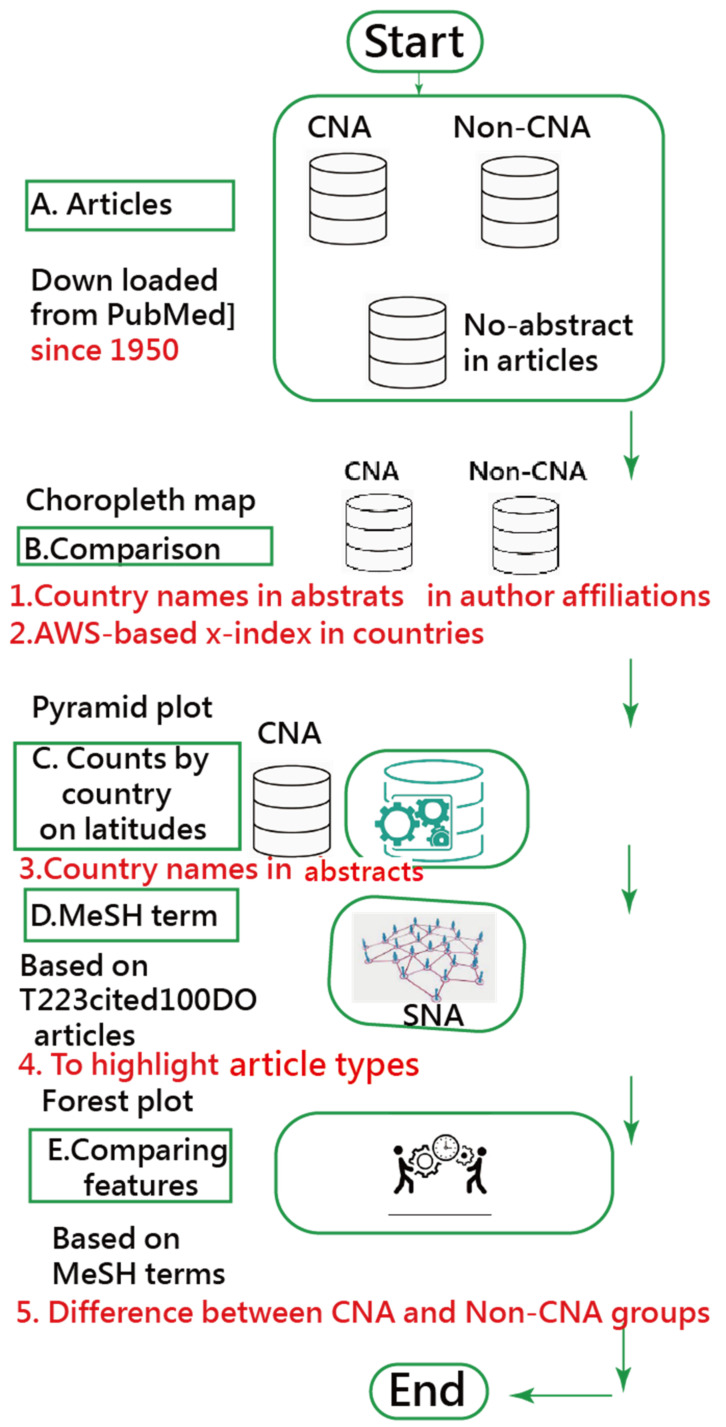
The study flowchart and contents. CNA, abstracts that included country names; Non-CNA, abstracts without country names; AWS, author-weighted scheme; SNA, social network analysis.

**Figure 2 ijerph-18-03197-f002:**
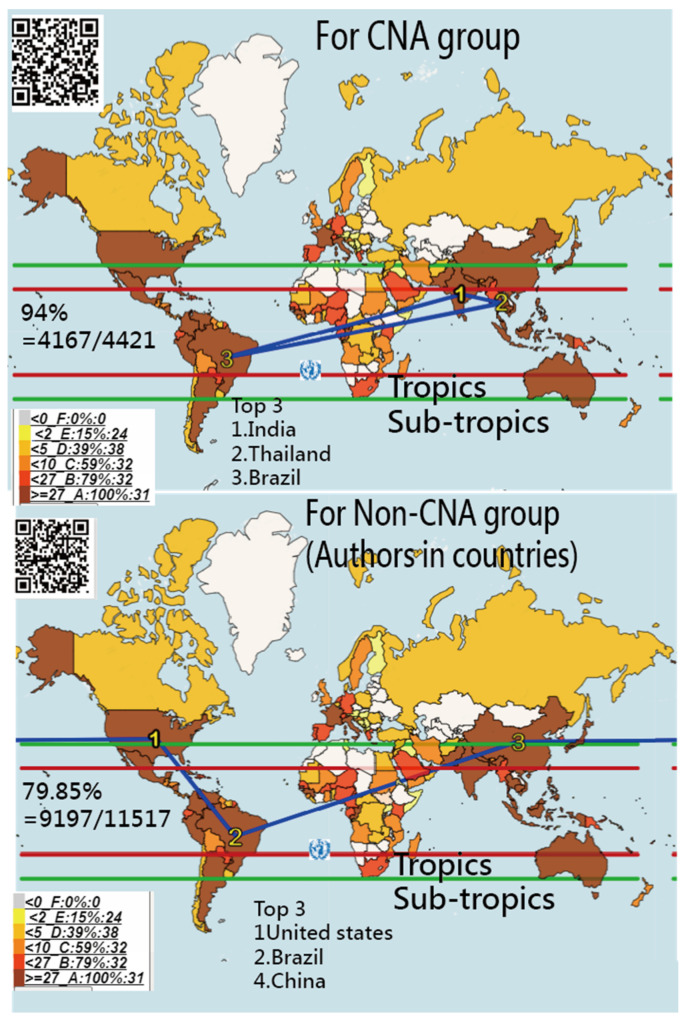
Count numbers of countries/areas in sub-tropics and tropics for the CNA (top) and Non-CNA (bottom) groups; the darker color means more numbers in collections.

**Figure 3 ijerph-18-03197-f003:**
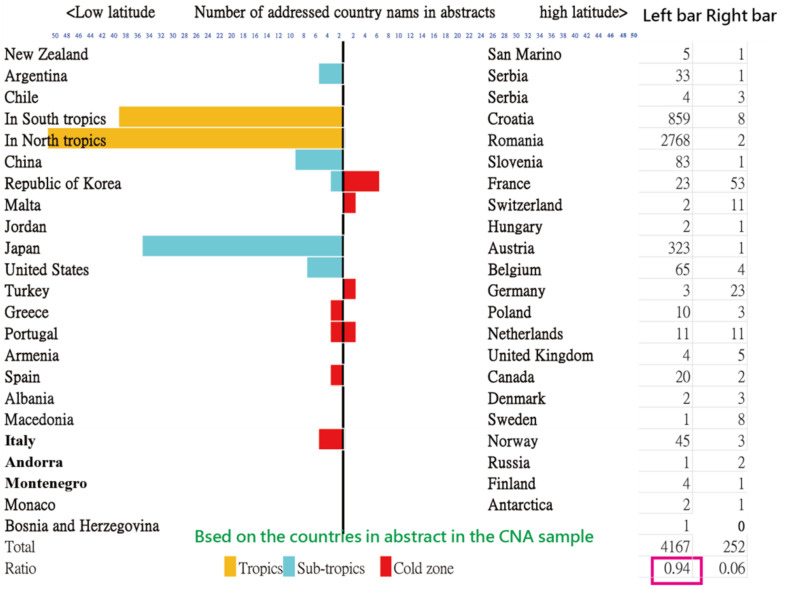
Distribution of dengue articles related to countries/areas; the latitudes of countries/areas are sorted in ascending order from top to bottom, and the bars are matched to the numbers in the columns at the right-hand side.

**Figure 4 ijerph-18-03197-f004:**
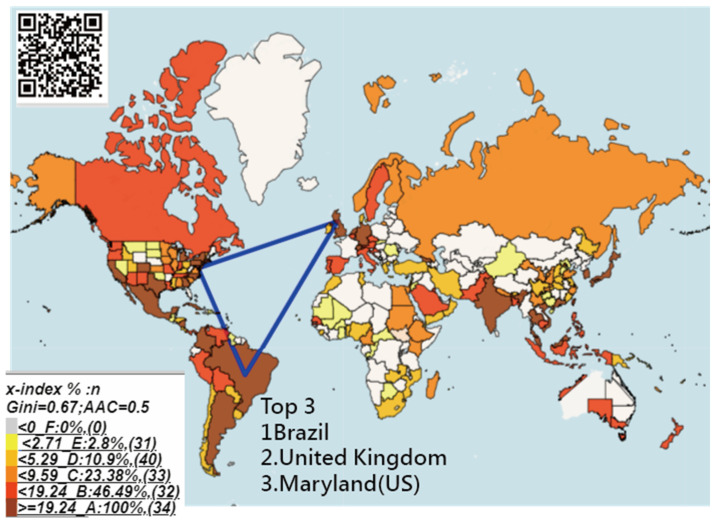
Research contributions as measured by x-indexes for articles regarding countries/areas in abstracts around the world; the darker means higher x-index and more contributions to the dengue discipline.

**Figure 5 ijerph-18-03197-f005:**
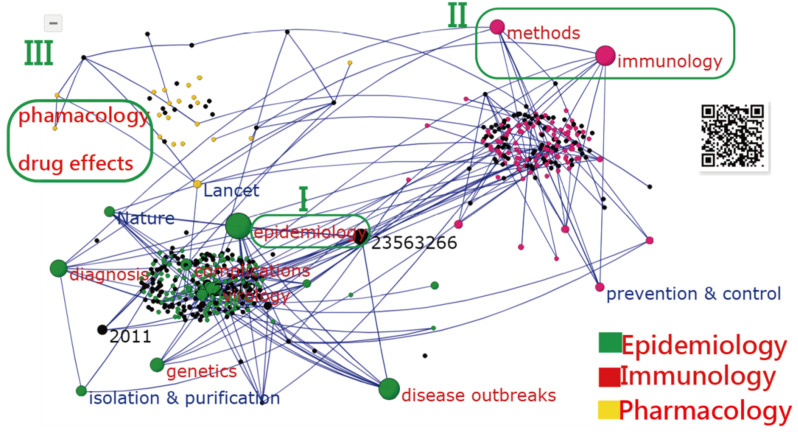
The most cited countries/areas and articles on the topic of dengue; three clusters are separated in the network, and the bubble sizes of the articles’ citations and centrality degree in SNA based on the top-cited articles with at least 100 citations (T100DO) articles.

**Figure 6 ijerph-18-03197-f006:**
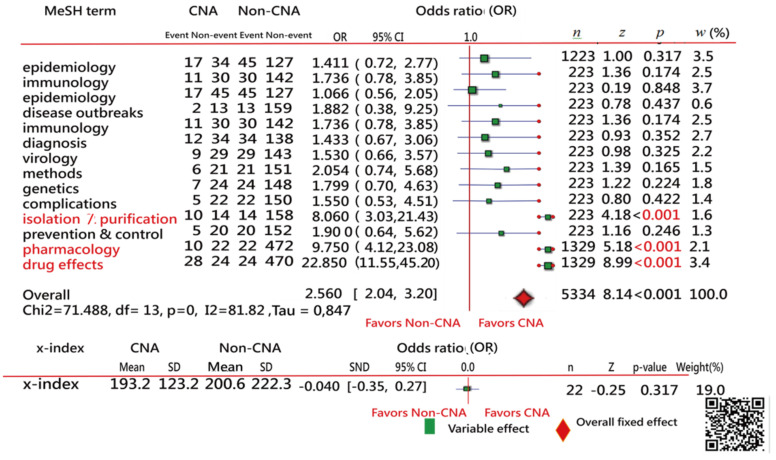
Comparison of features in the two samples of CNA and Non-CNA resulting in a forest plot; only three MeSH terms (in red) with different proportions (*p* < 0.05) on count events and non-events were found in articles between the two collections based on the T100DO articles.

**Table 1 ijerph-18-03197-t001:** Articles related to dengue outbreak (DO) outside the sub-tropical and tropical area (STTA).

No.	Country	Ref.	Patients From	MeSH Terms	
1	Finland	[32]	2016	Maldives	Dengue Virus; Travel
2	Russia	[33]	2017	Russia	Russia/epidemiology;
					Dengue */epidemiology
3	Norway	[34]	1997	Norway	Dengue/epidemiology;
					Norway/epidemiology
4	Sweden	[35]	2009	Thailand	Dengue/epidemiology;
					Sweden/epidemiology
5	Sweden	[36]	2005	Travelers	Dengue/diagnosis;
					Sweden
6	Denmark	[37]	2000	Travelers	Dengue */diagnosis;
					Travel
7	Canada	[38]	2013	United States	Dengue
8	France	[39]	1993	Thailand	Dengue/diagnosis;
					Thailand
9	Portugal	[40]	2016	Europe	Dengue/epidemiology
10	Italy	[41]	2014	India	Dengue/epidemiology;
					Disease Outbreaks

* denotes MeSH Major Topic in PubMed.

## Data Availability

All data were deposited in Appendix A.

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
