# Peer review of "A Bibliometric Analysis on Dengue Outbreaks in Tropical and Sub-Tropical Climates Worldwide Since 1950"

_ijerph, 2021, doi:10.3390/ijerph18063197_

Round 1
Reviewer 1 Report
Survey of the DENV literature, of great interest, well presented and carried out with care and scientific rigor. Deserves publication.
Author Response
Response: Thank a lot.

Reviewer 2 Report
Liu et al., described their analyzing results of the percentages of publications on Dengue outbreak in Tropical and Subtropical area using bibliometric analysis. Their results can provide the overall knowledge regarding dengue outbreak and the correlation between literature publications and dengue outbreak in the countries located in different climatic regions. However, there are still some questions which need further clarifications.
- In lines 41-46, it is not clear that why authors just mentioned and introduced about the status of autochthonous transmission and potential dengue outbreak in Japan in 2014 and 2020, respectively. How about the other countries ? not important ?
- In lines 57-61, authors described “There is little doubt that the DOs that occurred in sub-/tropical areas could be captured via capturing publications in Pubmed Central (PMC) for estimating the percentage (=n/N, n= the number of countries in sub-/tropics, N= the total number of countries with DOs)”. Please define and describe the motivation and formula more clearly and correctly.
- Although the authors described that “ Whether all DOs were in tropical and sub-tropical areas has not been verified before”, there were some similar publications which could be found from database, such as Virol J. 2016; 13: 78 ; FUTURE VIROLOGYVOL. 11, NO. 9; Glob Health Action. 2018;11(1):1504398. It is suggested that the novelty and difference between this work and other previous publications should be compared.
- Please explain the correlation and importance between dengue infected patients’ statistical reports from CDC or WHO and dengue publications in different countries located in subtropical and tropical regions. Because the importance of this study is not fully understood regarding analyses of how many publications in dengue outbreaks and the real situation of dengue epidemic status.
- In section, 3.2. Task 2: The DO Publications Beyond Sub-/Tropics. There were listed a lot of papers outside the sub-/tropics. Please summarize these papers in a Table for easy reading.
- In line 186 and Figure 5, the authors described “ The percentage of dengue alerts in the sub-tropics was 0.96(=14/350)”. Please confirm 0.96(=14/350) ?
- Please discuss the correlation between country development, economic and health medicine progression with literature publications of dengue outbreaks. Because some dengue outbreak countries did not pay attention to offer publications to report their dengue disease severity, leading to some bias in bibliometric analysis.
- The authors described that percentage of DO areas identical to the first-author affiliated countries account for 65.98%. However, currently, there are many international collaborations working together to fight dengue outbreak, leading to the first author or even the corresponding author is not correlated with that dengue outbreak country. How to resolve this problem in current analysis?
Author Response
Reviewer 2:
Liu et al., described their analyzing results of the percentages of publications on Dengue outbreak in Tropical and Subtropical area using bibliometric analysis. Their results can provide the overall knowledge regarding dengue outbreak and the correlation between literature publications and dengue outbreak in the countries located in different climatic regions. However, there are still some questions which need further clarifications.
- In lines 41-46, it is not clear that why authors just mentioned and introduced about the status of autochthonous transmission and potential dengue outbreak in Japan in 2014 and 2020, respectively. How about the other countries ? not important ?
Response: We have rewritten the paragraph following the previous sentence mentioned by the reviewer as below::
As such, the Japanese government is concerned about the DO and takes extra precaution against emerging infectious threats during the 2020(or 2021) Summer Olympics and Paralympics in Tokyo [8]. Japan partially locates in the sub-tropical climate, not to mention other countries with the whole territory in sub-tropical and tropical climates. We are motivated to investigate how many percentage of DOs exist in STTA.
- In lines 57-61, authors described “There is little doubt that the DOs that occurred in sub-/tropical areas could be captured via capturing publications in Pubmed Central (PMC) for estimating the percentage (=n/N, n= the number of countries in sub-/tropics, N= the total number of countries with DOs)”. Please define and describe the motivation and formula more clearly and correctly.
Response: The revised sentences have been rewritten as below:
We are in doubt whether the percentage of DOs in STTA could be obtained via capturing DO publications by computing the formula (=n/N, n= the number of documented countries with DO in STTA, N= the total number of DO countries in documents) instead of using the traditional surveillance, outbreak investigation, or other measures. We were interested in conducting bibliometric analyses to estimate the DO percentage in STTA.
- Although the authors described that “ Whether all DOs were in tropical and sub-tropical areas has not been verified before”, there were some similar publications which could be found from database, such as Virol J. 2016; 13: 78 ; FUTURE VIROLOGYVOL. 11, NO. 9; Glob Health Action. 2018;11(1):1504398. It is suggested that the novelty and difference between this work and other previous publications should be compared.
Response: The article mentioned by the reviewer at https://www.ncbi.nlm.nih.gov/pmc/articles/PMC6095018/#F0003 was just using the traditional bibliometric analysis instead of the percentage of Dos obtained from the published articles through two sources of (1) countries in abstracts and (2) not in abstract as we did in the manuscript.
- Please explain the correlation and importance between dengue infected patients’ statistical reports from CDC or WHO and dengue publications in different countries located in subtropical and tropical regions. Because the importance of this study is not fully understood regarding analyses of how many publications in dengue outbreaks and the real situation of dengue epidemic status.
Response: Thanks for giving us another opportunity to clarify the implication of DO percentage in STTA.
- In the revised manuscript. We have clarified the methodology of the two CNA and Non-CNA collections. From which, the DO percentage in STTA can be obtained by the mentioned countries in abstracts divided by the total mentioned countries in abstracts.
- Indeed, we do not guarantee that the assumption of countries in abstract must be DO occurred before in those countries. We illustrated the case of earthquake that was occurred in country nouns mentioned in articles.
- We found that a handful of countries that reported DO was because of travelers from the countries in STTA. If these countries in the cold zone(e.g., Norway and Russia) were excluded, the percentage of DO would be increased from 94.3% in our study to a higher level.
- Dengue infected patients’ statistical reports from CDC or WHO and dengue publications in different countries located in subtropical and tropical regions have been verified in many articles. The correlation and importance between the fact and the research from countries in documents is a clue that should be pursued.
(5) In our study, we used the text-mining techniques to extract countries in abstracts and verified the percentage of DO at 93.4% in STTA worthy of further investigations in the future.
- In section, 3.2. Task 2: The DO Publications Beyond Sub-/Tropics. There were listed a lot of papers outside the sub-/tropics. Please summarize these papers in a Table for easy reading.
Response: As advised by the reviewer, we have tabulated one Table to list the summary of those articles in Results 3.3.
- In line 186 and Figure 5, the authors described “ The percentage of dengue alerts in the sub-tropics was 0.96(=14/350)”. Please confirm 0.96(=14/350) ?
Response: In the revised manuscript, we have removed the original Figure 5 replaced with others to make the data cleanr and comprehensive than the original version.
.
- Please discuss the correlation between country development, economic and health medicine progression with literature publications of dengue outbreaks. Because some dengue outbreak countries did not pay attention to offer publications to report their dengue disease severity, leading to some bias in bibliometric analysis.
Response: As advised by the reviewer, one paragraph in limitation has been raised and emphasized on the issue in the revised version of manuscript.
- The authors described that percentage of DO areas identical to the first-author affiliated countries account for 65.98%. However, currently, there are many international collaborations working together to fight dengue outbreak, leading to the first author or even the corresponding author is not correlated with that dengue outbreak country. How to resolve this problem in current analysis?
Response: In the revised version, we have redone the analysis updating data up to 10 March 2021. The oercentage of DO areas has been increase to 79.85%(see the bottom panel in Figure 2) when all co-authors have been taken into account of matching the affiliated countries with the STTA..

Reviewer 3 Report
Reviewer Comments,
This paper described a bibliometric analysis of Dengue Outbreaks in tropical and sub-tropical regions from Worldwide since 1950. The study design and methodology is sound for some extent, but may be improved to increase its clarity. The results are well-presented.
Comments:
- Please improve the language aspects of the manuscript as there are grammatical errors and typos.
- Please consider changing the title to: A Bibliometric Analysis on Dengue Outbreaks in Tropical and Sub-Tropical Climates Worldwide Since 1950.
- Suggestion: In discusión could include the findings made by Zyoud, 2016 (Dengue research: a bibliometric analysis of worldwide and Arab publications during 1872–2015), Satish and Vasna, 2019 (Scientific research publications in dengue: A global and Indian bibliometric analysis from 1997 to 2018), and Bhardwaj, 2014 (Dengue Fever: A Bibliometric Analysis of India’s Contributions to the Research Literature of This Dangerous Tropical Disease).
Author Response
Reviewer 3:
This paper described a bibliometric analysis of Dengue Outbreaks in tropical and sub-tropical regions from Worldwide since 1950. The study design and methodology is sound for some extent, but may be improved to increase its clarity. The results are well-presented.
Comments:
- Please improve the language aspects of the manuscript as there are grammatical errors and typos.
Response: In the revised version, we have made improvement in grammatical errors and typos throughout all the contexts.
- Please consider changing the title to: A Bibliometric Analysis on Dengue Outbreaks in Tropical and Sub-Tropical Climates Worldwide Since 1950.
Response: As advised by the reviewer, the article title has been replaced.
- Suggestion: In discusión could include the findings made by Zyoud, 2016 (Dengue research: a bibliometric analysis of worldwide and Arab publications during 1872–2015), Satish and Vasna, 2019 (Scientific research publications in dengue: A global and Indian bibliometric analysis from 1997 to 2018), and Bhardwaj, 2014 (Dengue Fever: A Bibliometric Analysis of India’s Contributions to the Research Literature of This Dangerous Tropical Disease).
Response: As advised by the reviewer, we have added these references in Discussions.

Reviewer 4 Report
In this study, Liu et al. evaluated publications on dengue outbreaks (DOs) since 1950 and defined the percentage of publications in the tropical and subtropical areas; outside the sub-/tropics, DOs per country, and the most cited articles on DOs. In general, the study is reasonable but challenging to follow.
I have some recommendations that can improve the manuscript:
I would suggest being specific on the date range of the articles included (e.g., January 1950 to December 2020) and whether all the articles were in English, and any other relevant information.
In my opinion, the figure's legend should be descriptive (it should explain the figure itself). You have to add the meaning of the box in Figures 1 and 3.
Figure 2. It is confusing and difficult to interpret. You may clarify what the bars represent as well as the numbers on the right.
Figure 4. From 10 bubbles that I clicked randomly, I was able to see only four articles. Please consider reviewing the figure.
In my opinion, the introduction and result sections need to be improved.
Line 177. I wonder whether the authors could provide the date of citation consultations. Those citations do not correlate with the ones found in Scopus (1136) or google academic (1748) for PMID 10608744, for example. I agree that you added that limitation in the discussion.
Line 45. Is the slash correct in sub-/tropical (beginning of the sentence)?
There are several missing spaces (e.g., lines 86, 89).
Line 89. I did not see any additional files.
Line 99. Which abstract 1?
Line 202. Correct DNT to NTD
Author Response
Reviewer 4:
In this study, Liu et al. evaluated publications on dengue outbreaks (DOs) since 1950 and defined the percentage of publications in the tropical and subtropical areas; outside the sub-/tropics, DOs per country, and the most cited articles on DOs. In general, the study is reasonable but challenging to follow.
I have some recommendations that can improve the manuscript:
I would suggest being specific on the date range of the articles included (e.g., January 1950 to December 2020) and whether all the articles were in English, and any other relevant information.
Response: We have redone the analysis and downloaded from PMC from January 1950 to December 2020 in the revised manuscript. .
In my opinion, the figure's legend should be descriptive (it should explain the figure itself). You have to add the meaning of the box in Figures 1 and 3.
Response: Those Figures have been reproduced in the revised manuscript according to the reviewer’s advice. .
Figure 2. It is confusing and difficult to interpret. You may clarify what the bars represent as well as the numbers on the right.
Response: We have reproduced the Figure and clarified the instructions in the Figure legend.
Figure 4. From 10 bubbles that I clicked randomly, I was able to see only four articles. Please consider reviewing the figure.
Response: This Figure has been reproduced. Only the black bubble can be available for linking to the article in PubMed.
In my opinion, the introduction and result sections need to be improved.
Response: As suggested by the reviewer, we have made a lot of changes in the revised version.
Line 177. I wonder whether the authors could provide the date of citation consultations. Those citations do not correlate with the ones found in Scopus (1136) or google academic (1748) for PMID 10608744, for example. I agree that you added that limitation in the discussion.
Response:
- We have added the date of data extraction from Pubmed in the sentence as below:
We downloaded 8922 abstracts by searching keywords “dengue[MeSH Major Topic]” from PMC since 1950 as of 10 July 2019.
- The article(=PMID=10608744) has 562 citations as of 10 March 2021(https://pubmed.ncbi.nlm.nih.gov/10608744/) , significantly less than the citations in Scopus (1136) or google academic (1748). That is because the latter two involve more irrelevant original research in the databases. Importantly, the original articles with peer-review process were taken into this study.
Different databases might result in disparate number of citation for a given article; see the article entitled “Comparison of PubMed, Scopus, Web of Science, and Google Scholar: strengths and weaknesses “ at https://pubmed.ncbi.nlm.nih.gov/17884971/ .
Line 45. Is the slash correct in sub-/tropical (beginning of the sentence)?
Response: We have replaced all such term of “sub-/tropical” with “subtropical and tropical” in the whole context.
There are several missing spaces (e.g., lines 86, 89).
Response: The spaces have been added into the sentences.
Line 89. I did not see any additional files.
Response: We replaced the additional files with Appendix 1 because all additional files are deposited in the link at https://osf.io/9jasg/?view_only=3dcfd9e7648748f6aa5cd0aaedf30149
Line 99. Which abstract 1?
Response: We have revised the sentence as below:
-à After calculating the percentage mentioned in 2.2 Task 1(i.e., of the publications on DOs in sub-/tropics using text-mining techniques after extracting names based on counties/areas in abstracts). we listed several typical DO 95 documents that were published outside the sub-/tropics.
Line 202. Correct DNT to NTD
Response: The typo has been corrected.

Reviewer 5 Report
In this study the authors to describe How Many Percentages of Publications on Dengue Outbreaks Were in Tropical and Sub-Tropical Climates Worldwide Since 1950. The study is interesting, however, there are observations in the selection of the included studies. It is only limited to PMC.
Bear in mind that not all DOs in each country have been published in journals.
Review the results and conclusion
The search strategy is not clear.
Why did you download articles only from PMC? Authors should conduct a systematic search using, MEDLINES, SCOPUS, Web of Science, etc.
How have articles on dengue outbreaks been selected?
What were the inclusion and exclusion criteria?
What types of articles have you included? It is not possible to appreciate how many are articles of originals, reviews, systematic reviews.
The most design for conducting a literature search is to follow the PRISMA guidelines.
Was there a peer review to select articles on dengue outbreaks?
How did you define a dengue outbreak to select an item?
I suggest include the PRISMA algorithm.
How they have differentiated 2 or more articles of the same outbreak in a country.
Author Response
Reviewer 5:
In this study the authors to describe How Many Percentages of Publications on Dengue Outbreaks Were in Tropical and Sub-Tropical Climates Worldwide Since 1950. The study is interesting, however, there are observations in the selection of the included studies. It is only limited to PMC.
Response: In Limitation 1, we have addressed that the difference in results might exist due to disparate databases used in bibliometric analysis.
Bear in mind that not all DOs in each country have been published in journals.
Response: The study aim is to compute the percentage of DO in STTA. As such, the absence of DO publications would not largely change the percentage. That is, only when the presence of DO publications will increase the percentage of DO in STTA and the percentage will be decreased if countries outside the STTA appear in DO articles.
Review the results and conclusion
The search strategy is not clear.
Why did you download articles only from PMC? Authors should conduct a systematic search using, MEDLINES, SCOPUS, Web of Science, etc.
Response: This study is bibliometric analysis instead of Meta-analysis. As such, only one database is used to analyze articles. Otherwise, citation analysis would be problematic due to different citation in different database. If one articles is in two or more databases, the problem of which citations should be adopted in analysis will be encountered; see the link at https://link.springer.com/article/10.1007/s11301-020-00188-4
.
How have articles on dengue outbreaks been selected?
Response: In the revised manuscript, we have added Figure 1 to explain the study flowchart. A MP4 video is provided in Appendix 1 which will help readers capture the content of analysis in this study.
The articles were extracted from PMC by searching keywords “dengue[MeSH Major Topic]”, see the link at https://osf.io/2cpfb/
What were the inclusion and exclusion criteria?
Response: In the revised version, we divided collections into three samples of CNA, Non-CNA, and no abstract in articles. Only the former two were included in this study because comparison was made in percentage of DO in STTA between the two collections. .
What types of articles have you included? It is not possible to appreciate how many are articles of originals, reviews, systematic reviews.
Response: Only abstracts in articles were included. As such, all those articles including originals, reviews, systematic reviews. In the revises manuscript, we further analyze data using the T100DO articles(i.e., at least cited 100 times) shown in Appendix 3 at https://osf.io/9jasg/?view_only=3dcfd9e7648748f6aa5cd0aaedf30149
.
The most design for conducting a literature search is to follow the PRISMA guidelines.
Response: Usually the PRISMA guidelines is provided to the study of Meta-analysis. Few studies of bibliometric analysis provide the PRISMA diagram and checklist to the articles. That is because usually more tables and Figures are required to present in bibliometric analysis than the Meta-analysis. The article space is not allowed to provide additional PRISMA diagram and checklist to the given article.
Was there a peer review to select articles on dengue outbreaks?
Response: The only inclusion criterion is the dengue-related articles. Because the MeSH terms have been added to the articles, additional review the article is not necessary.
Medical Subject Heading (MeSH) in PubMed was used to get specific dengue related documents. MeSH is a controlled vocabulary created by the NLM to index bibliographical archives. Experienced indexers in NLM perform standard steps and procedures before assigning MeSH terms into any new article in Pubmed; see the medline indexing process: determining subject content. [cited 2018. July 13]. Available at https://www.nlm.nih.gov/bsd/disted/meshtutorial/principlesofmedlinesubjectindexing/theindexingprocess/.
How did you define a dengue outbreak to select an item?
I suggest include the PRISMA algorithm.
Response: we do not guarantee that the assumption of countries in abstract must be DO occurred before in those countries. We illustrated the case of earthquake that was occurred in country nouns mentioned in articles. As such, a dengue-related article reporting a specific country/area in abstract can be ensured that the country/area has a extremely high probability to be the DO area. Otherwise, we did not image any reason that a county noun (instead of adjective) in abstract is not related to the outbreak area. ,
How they have differentiated 2 or more articles of the same outbreak in a country.
Response: The aim of this study is to computer the percentage of DO in STTA. Even if the similar DO in many articles that will not change the fact of the area located within the SSTA or beyond the SSTA.

Round 2
Reviewer 2 Report
All the comments were addressed properly.
Author Response
Thanks

Reviewer 5 Report
No comments
Author Response
Thanks
